# Estrogenic Modulation of Ionic Channels, Pumps and Exchangers in Airway Smooth Muscle

**DOI:** 10.3390/ijms24097879

**Published:** 2023-04-26

**Authors:** Bianca S. Romero-Martínez, Bettina Sommer, Héctor Solís-Chagoyán, Eduardo Calixto, Arnoldo Aquino-Gálvez, Ruth Jaimez, Juan C. Gomez-Verjan, Georgina González-Avila, Edgar Flores-Soto, Luis M. Montaño

**Affiliations:** 1Departamento de Farmacología, Facultad de Medicina, Universidad Nacional Autónoma de México, Ciudad de México 04510, Mexico; 2Laboratorio de Hiperreactividad Bronquial, Instituto Nacional de Enfermedades Respiratorias “Ismael Cosío Villegas”, Ciudad de México 14080, Mexico; 3Neurociencia Cognitiva Evolutiva, Centro de Investigación en Ciencias Cognitivas, Universidad Autónoma del Estado de Morelos, Cuernavaca 62209, Mexico; 4Departamento de Neurobiología, Dirección de Investigación en Neurociencias, Instituto Nacional de Psiquiatría “Ramón de la Fuente Muñiz”, Ciudad de México 14370, Mexico; 5Laboratorio de Biología Molecular, Departamento de Fibrosis Pulmonar, Instituto Nacional de Enfermedades Respiratorias Ismael Cosío Villegas, México City 14080, Mexico; 6Laboratorio de Estrógenos y Hemostasis, Facultad de Medicina, Universidad Nacional Autónoma de México, Ciudad de México 04510, Mexico; 7Dirección de Investigación, Instituto Nacional de Geriatría (INGER), Ciudad de México 10200, Mexico; 8Laboratorio de Oncología Biomédica, Instituto Nacional de Enfermedades Respiratorias “Ismael Cosío Villegas”, México City 14080, Mexico

**Keywords:** estrogen, airway smooth muscle, channels, exchangers, pumps, receptors

## Abstract

To preserve ionic homeostasis (primarily Ca^2+^, K^+^, Na^+^, and Cl^−^), in the airway smooth muscle (ASM) numerous transporters (channels, exchangers, and pumps) regulate the influx and efflux of these ions. Many of intracellular processes depend on continuous ionic permeation, including exocytosis, contraction, metabolism, transcription, fecundation, proliferation, and apoptosis. These mechanisms are precisely regulated, for instance, through hormonal activity. The lipophilic nature of steroidal hormones allows their free transit into the cell where, in most cases, they occupy their cognate receptor to generate genomic actions. In the sense, estrogens can stimulate development, proliferation, migration, and survival of target cells, including in lung physiology. Non-genomic actions on the other hand do not imply estrogen’s intracellular receptor occupation, nor do they initiate transcription and are mostly immediate to the stimulus. Among estrogen’s non genomic responses regulation of calcium homeostasis and contraction and relaxation processes play paramount roles in ASM. On the other hand, disruption of calcium homeostasis has been closely associated with some ASM pathological mechanism. Thus, this paper intends to summarize the effects of estrogen on ionic handling proteins in ASM. The considerable diversity, range and power of estrogens regulates ionic homeostasis through genomic and non-genomic mechanisms.

## 1. Introduction

Homeostasis is a state of cellular equilibrium maintained through regulatory mechanisms resistant to change that preserve the constant physiological conditions. The composition of the cytosol, the optimal internal cellular environment, differs significantly from that of the extracellular space, and these spaces are separated by a semipermeable lipid bilayer called cellular membrane that functions not only as a barrier between these two spaces but also prevents charged particles from crossing freely from one side to the other. Several cellular functions, as for instance the action potential in the nerves, depend on ions permeating through the cellular membrane. Conceivably, the ionic composition on either side of the cell membrane is different and many cellular mechanisms work in accordance to maintain a constant concentration of them and consequently, an electrochemical gradient [1,2]. In order to maintain this balance, cells express a variety of proteins embedded in the cell membrane and in organelle membranes that function as gatekeepers of ions. Some of these proteins transport the ions in and out of these membranes in a highly controlled manner [1,2,3]. These proteins can be classified into three categories: ionic channels, pumps, and exchangers (Figure 1) [1,2].

Protein ion channels allow specific ions to flow across the cell membrane in favor of their electrochemical gradient, consequently generating transmembrane electric currents [3]. Ion channels are widely diverse, with several hundred genes in the human genome encoding them. Yet, they share certain structural generalities that characterize them, such as, for instance, being composed of several transmembrane segments, with a specific region that forms a pore across the membrane that serves as the passageway for the ions to cross at high speed. The channels dynamically alternate between an open or closed state through a “gating” process regulated by other segments of the channel that allow the flow of ions when needed [3,4]. In some cases, some ion channels may be in an inactive state, a circumstance that keeps them closed even during a process of resting membrane potential change. Therefore, ion channels are essential for the development of electrophysiological responses of excitable cells.

On the other hand, ion pumps do require a source of energy achieved through the hydrolysis of ATP, to move the ions against their electrochemical gradient; this process is referred to as active transport [3]. Pumps also utilize a gating system to regulate the passage of ions through their pore. However, unlike channels, pumps have a two-gate system that alternates between open and closed states. Gating reactions requiring ATP are needed for the gates to alternate between open and closed states, limiting the speed of uphill movement of the ions [3].

The third type of transporters are exchangers. They transfer different ions in or out of the cytosol simultaneously, they do not require ATP as an energy source, and instead exploit the electrochemical gradient by moving the ions downhill of the concentration gradient. This type of transport is also known as facilitated transport. Three types of exchangers have been identified. Uniporters transport a single type of ion in one direction in favor of the concentration gradient. The second type are symporters, where two different types of ions are moved simultaneously across the membrane in the same direction. The last type are called antiporters, proteic structures apt to move different types of ions across the membrane in opposite directions (Figure 1) [3,5].

To maintain ionic homeostasis and provide a stable electrophysiological status in cells, these proteins are strictly regulated by numerous pathways, both extrinsic (autocrine, paracrine, endocrine signaling) and intrinsic cellular mechanisms and the electrochemical gradients [6]. Ion handling proteins can be regulated acutely (modulating its activity), or chronically (through modulation of quantity and subtype expression of the proteins). Acute regulation of ion handling proteins can be through direct interaction of the protein with the regulatory molecule, or indirectly, through second messengers and protein kinases [4]. Chronic regulation determines the number and kinds of ion transporters expressed in any given cell and requires three stages of regulation: developmental, homeostatic, and evolutionary [6]. Developmental regulation involves the activation of large quantities of genes necessary to express a specific cellular phenotype; these are tissue-specific patterns. Therefore, the quantity and subtypes of transporters expressed by any given cell type will vary from any other type. This expression of a defined genetic pattern certainly defines the cellular phenotype that best serves its function [6]. Meanwhile, homeostatic regulation refers to two main processes. The first keeps a certain phenotypic expression stable under physiological conditions, and the second oversees the plasticity of the transporter’s expression in response to changes in the physiological stimuli [6].

In this sense, hormones have been recognized as one of the main homeostatic regulatory systems, and in recent years sex steroid hormones (SSHs) gained increasing interest for their role outside of reproductive physiology. SSHs have been implicated in regulating many critical physiological processes along every stage of life, including growth, regulation of homeostasis, reproduction, and aging [7,8]. Specifically, in lung physiology, SSHs participate in developmental stages from embryonic and onwards [9,10,11]. In women, significant hormonal changes occur during their reproductive life, initiating during puberty and lasting until menopause. Most of these changes are regulated through estrogens, SSHs that belong to a group of aromatized 18-carbon steroids whose induction and maintenance of various physiological processes makes them fundamental in reproduction, development of behaviors and several non-sexual processes [12]. Among the many organ systems in which estrogens exert their effect, the cardiovascular system has been abundantly studied, particularly vascular smooth muscle and cardiac muscle and the role estrogens play on their ion transporter regulation [13,14,15]. Many similarities in the activity of estrogen can be observed between vascular and airway smooth muscle (ASM), and owing to the tremendous amount of research dedicated to vascular diseases, it is quite enticing to apply the findings to ASM. However, key differences exist between these two cell types, and although year after year the research in the field of the role of SSHs in ASM grows, this topic still has much room for further investigation [11].

Here, we review emerging evidence suggesting estrogens influence airway smooth muscle function at the cellular level, modifying processes involved in intracellular ion homeostasis on different physiological and pathological conditions.

## 2. Estrogens Biosynthesis and Modes of Action

Estrogens belong to the family of SSHs derived from cholesterol, and they share a common structural base consisting of four rings called cyclopentanoperhydrophenanthrene [7,8,12]. In women, their biosynthesis mainly occurs in the ovaries. However, extragonadal sites have been identified as, for instance, the adrenal glands, liver, heart, skin, brain, adipose tissue and lung, in which an aromatase enzyme is expressed; this enzyme also functions as the regulating step in estrogen production [7,8,12,16,17,18]. Local biosynthesis of estrogens can have tissue-specific effects that could participate in physiological processes and induction or maintenance of various diseases (Figure 2) [12,16].

During the reproductive lifetime of women, three main estrogens are physiologically synthesized: estrone (E1), 17β-estradiol (E2), and estriol (E3); E2 has the highest production representing the most abundant and potent estrogen during a woman’s reproductive lifetime. In non-pregnant women, physiological circulating levels ranging from 80 pM–1.5 nM and going up to 150 nM during pregnancy [11,12]. In postmenopausal women, estradiol levels diminish, ranging between 40–120 pM [11].

Estrogens exert regulatory actions via long-term genomic and acute non-genomic actions [12,19]. Classical estrogen signaling (genomic effects) is mediated through the nuclear receptors ERα and ERβ, abundantly expressed in human ASM cells [11,12,19,20]. When ERs are activated by estrogen binding, they can lead to different signaling pathways, including gene transcription regulation either directly or through estrogen response elements (EREs) in the promoter region. ERs can also interact with other DNA-binding transcription factors and induce rapid effects involving phosphorylation processes [11,12,19]. Another estrogen receptor located at the cell membrane was identified. The activity of GPR30 can lead to the activation of numerous intracellular signaling pathways, including cyclic nucleotides, protein kinase C, protein kinase A, and protein kinase G [11,12,19]. However, it was demonstrated that the GPR30 was not significantly expressed in human ASM cells [21]. On the other hand, estrogens’ non-genomic effects are rapid and take place outside the cell nucleus and, as stated, can be initiated through estrogen membrane receptors. Another proposed mechanism of action is directly binding to its target protein [22]. Through these numerous mechanisms of action, estrogens are able to regulate ion transporters in ASM cells (Figure 2).

## 3. Airway Smooth Muscle Calcium Handling Mechanisms and Estrogens

Cell calcium (Ca^2+^) homeostasis is maintained by a finely tuned Ca^2+^ signaling system made-up of numerous Ca^2+^ transporters (channels, exchangers, and pumps) regulating the influx and efflux of this cation from the cytoplasm to preserve its balance. Ca^2+^ homeostasis is essential for the cell. As a second messenger, Ca^2+^ signaling regulates various cellular processes that depend on the time of the Ca^2+^ and concentration. It is well known that Ca^2+^ regulates exocytosis, contraction, protein phosphorylation, dephosphorylation, metabolism, gene transcription, fecundation, cell proliferation, and even apoptosis [23]. In the ASM, Ca^2+^ homeostasis keeps intracellular basal Ca^2+^ concentrations (b[Ca^2+^]i) at around 100–150 nM, while Ca^2+^ concentrations in the intracellular stores and extracellular space are higher (5–10 mM and 2 mM, respectively) creating a large chemical gradient in favor of Ca^2+^ influx into the cytosol [23,24]. In order to regulate [Ca^2+^]i, numerous proteins exists to facilitate the cellular influx and efflux of Ca^2+^. Among the calcium-handling proteins, we can include the voltage-dependent Ca^2+^ channels (VDCCs), store-operated Ca^2+^ channels (SOCCs), receptor-operated Ca^2+^ channel (ROCCs), transient receptor potential channels (TRPs), and the Na^+^/Ca^2+^ exchanger in its reverse form (NCX_REV_) as influx mechanisms located in the cellular membrane. On the other hand, the Na^+^/Ca^2+^ Exchanger (NCX) and the plasma membrane Ca^2+^ ATPase (PMCA) are efflux mechanisms located in the cellular membrane. Meanwhile, the sarcoplasmic reticulum (SR) functions as the main internal Ca^2+^ store in the ASM. On the SR membrane, we can find the inositol 1,4,5-trisphosphate receptor (IP_3_R) and the ryanodine receptors (RyR) as Ca^2+^ efflux mechanisms. In contrast, the sarcoplasmic reticulum Ca^2+^ ATPase (SERCA) acts as an intracellular Ca^2+^ extrusion mechanism (Figure 3) [23,24].

### 3.1. Voltage-Dependent Ca^2+^ Channels

Extracellular Ca^2+^ entry is primordial in Ca^2+^ signaling, and its influx is primarily driven by an electrochemical gradient. In ASM, both L- type VDCCs (Long Lasting Currents voltage-dependent Ca^2+^ channels) and T-type VDCCs (Transient Currents voltage-dependent Ca^2+^ channels) are expressed, but L-VDCC is the predominant type in human ASM and various other species [24,25,26,27,28,29,30,31,32,33]. In the sense, the preincubation with E2 at physiological levels can inhibit L-VDCCs non-genomically (Figure 3) in a concentration-dependent manner in human ASM (hASM) stimulated with histamine, with a more pronounced effect when using an ERα-selective agonist, which was not observed when using an ERβ selective agonist (Table 1) [20]. This could be partially due to the localization of the receptors since ERβ is minimally present in the plasma membrane, where non-genomic effects could take place [20]. This effect seems to have a biphasic response since at supraphysiological levels, E2 also inhibits L-VDCCs non-genomically in guinea pig ASM, as demonstrated through electrophysiological studies (Table 1) [34]. Both estrogen receptors seem to have different signaling pathways. They serve distinct purposes, both in physiological and pathological conditions. For instance, in human ASM cells from an asthmatic, an increase in the expression of the different variants of both receptors at different degrees has been reported, although their role in asthma pathophysiology remains to be elucidated [21]. Furthermore, chronic exposure (24 h, genomic effect) to estrogens in hASM cells from asthmatics and non-asthmatics seems to have opposing effects on [Ca^2+^]i. This phenomenon seems to depend on the type of receptor activated. An ERα-specific agonist ((R,R)-THC) augmented the [Ca^2+^]i response induced by histamine in both asthmatic and non-asthmatic hASM cells. Meanwhile, an ERβ-specific agonist (DPN), decreased the [Ca^2+^]i response induced by histamine in asthmatic and non-asthmatic hASM cells. The effect observed on the [Ca^2+^]i response with activation of ERβ signaling appears to implicate the inhibition of L-VDCC (Figure 3) (Table 1) [35]. Interestingly, E2, (as a non-selective ER agonist) at physiological concentrations did not show significant changes in the [Ca^2+^]i response to different agonists in ASM [35]. Unfortunately, the genomic effects of estrogen on L-VDCC expression, have not been explored in ASM cells yet. In comparison, ovariectomy (OVX) induced an increase in the channels’ expression in rat aorta. Interestingly, treatment with E2 downregulated the channels’ mRNA expression, and treatment with E2 and tamoxifen (an ER blocker) had a similar effect as E2 alone [36].

### 3.2. Store-Operated Calcium Channels

Another group of calcium channels that regulates Ca^2+^ influx to the cytosol is SOCCs, considered non-selective cation channels. Its activity depends on the SR Ca^2+^depletion and its goal is to contribute to the refilling of internal Ca^2+^ stores [37]. To carry out their function following SR depletion, these channels must assemble into a complex formed by two proteins: Orai1 (calcium release-activated calcium channel protein 1) and STIM1 (stromal interacting molecule) [37]. When at rest, STIM1 is bound to Ca^2+^. When Ca^2+^ levels begin to diminish in the SR, STIM1 disassociates from Ca^2+^, and this change causes STIM1 molecules to cluster and translocate to a region in proximity to the plasma membrane [37,38,39]. Orai1 is a transmembrane protein located in the plasma membrane. In basal conditions, this protein is a dimer, but when STIM1 clusters are formed, they interact with Orai1 and enable them to form tetramers, forming selective Ca^2+^ pores that allow Ca^2+^ influx [37,39]. This mechanism is negatively regulated by the SR transmembrane protein SARAF (store-operated Ca^2+^ entry-associated regulatory factor). When SARAF interacts with STIM1, it prevents spontaneous activation or the interaction between STIM1-Orai1 [37,40,41]. On the other hand, transient receptor potential canonical (TRPC) channels also play an important role in Ca^2+^ homeostasis, and it is known that the TRPC3 isoform prevails in ASM cells [24]. Recently, the STIM1-Orai1 complex was found to interact with TRPC channels required for their activation [42,43]. Other members of the TRP channel family are vanilloid (TRPV) receptors, ankyrin (TRPA) and melastatin (TRPM), of which TRPV1 and TRPV4 have been identified in ASM cells (Figure 3) [24,44,45,46,47].

Given the importance of SOCCs in Ca^2+^ homeostasis, the effects that E2 could have on their regulation is of increasing interest. Townsend et al. determined that the decrease in Ca^2+^ induced by the acute exposure to physiological concentrations of E2 observed in the Ca^2+^ response induced by histamine in hASM cells (Table 1) [20], was in part due to inhibition of SOCCs observed as a diminished SR refilling (Figure 3) [48]. On the other hand, the chronic effects of the estrogen receptor signaling on SOCCs modulation have also been explored in healthy and asthmatic hASM cells. The chronic exposure (24 h) to an ERβ-selective agonist (WAY-200070) downregulated the expression of STIM1 and Orai1 measured through Western blot (Figure 3), and consequently produced a decrease in [Ca^2+^]i. This effect was observed in hASM from non-asthmatics, but was more pronounced in asthmatics [49]. When exposed to an ERα-selective agonist (propyl pyrazole triol at 10 nM) for 24 h, an opposite effect was observed. SOCCs Ca^2+^ influx was increased, and the expression of STIM1 and Orai1 was also increased (Figure 3) (Table 1) [49].

Cigarette smoke (CS) is a common risk factor associated with many airway diseases, and asthmatic women exposed to CS tend to have a worse asthmatic response; therefore, the effects of CS and E2 on ASM Ca^2+^ regulation were explored [50]. As a result, it was defined that, in hASM cells, a 24 h exposure to CS extract induces a significant increase in the Ca^2+^ response to histamine. On the other hand, acute exposure to nanomolar E2 concentrations inhibits the Ca^2+^ response to histamine partially via inhibition of SOCCs [20]. When exposed to CS extract, this E2 effect was blunted [50]. Chronic exposure to CS extract (24h exposure to 1% or 2% CS) also seems to increase the expression of ERα and ERβ, leading to investigate if a differential ER regulation was present. As observed before, the acute effect on Ca^2+^ was present when using an ERα-specific agonist but not an ERβ-selective agonist [20]. This effect was absent when the cells were previously exposed to CS extract [50]. Therefore, CS extract enhances [Ca^2+^]i through the dysregulation of ER signaling, and blunts the acute reduction in [Ca^2+^]i and subsequent force generation resultant from ERα activation, more so in asthmatic patients than in healthy subjects. E2 also non-genomically inhibited STIM1 phosphorylation, while pre-exposure to CS extract for 24 h abolished this E2 effect [50].

Studies in other cellular types point out the physiological implications of E2-mediated regulation of SOCCs. In airway epithelial cells, E2 (10 nM) acute (15 min) exposure can reduce STIM1 phosphorylation, preventing the formation of STIM1 clusters from interacting with and activating Orai1, decreasing SOCCs activity. This decrease in SOCCs influx also affects Ca^2+^ activated Cl^−^ channels colocalized with Orai1, impacting mucus hydration [51]. In mouse embryonic stem cells (mESCs), the treatment with physiological concentrations of E2 (1 pM and 1nM) during 24 h enhanced cellular proliferation in a concentration-dependent manner. The effect was through SOCCs activity and could be reverted by SOCCs blockers (2-APB 0.3 µM) [52]. As already mentioned, Ca^2+^ homeostasis disruption can lead to pathologies. Such is the case in various cancer types, where dysfunction of Ca^2+^ homeostasis has been implicated. Epithelial ovarian cancer (EOC) is closely tied to E2 regulation, and the effects that E2 could have over Orai1 and the different pathological processes in EOC have been explored [53]. After 12 h exposure with E2 (at a micromolar range), Orai1 expression increased in SK-OV-3 cells (an EOC cell line), with the most significant effect observed at 1 µM, leading to an increase in [Ca^2+^]i. E2, through Orai1, positively regulated cellular and migration (by CDK6 and MMP-1 pathways, respectively), and suppressed cellular proliferation apoptosis through caspase3 expression regulation [53]. In another study in EOC cells, the chronic exposure to E2 (10 nM–1 µM for 24, 48, and 72 h) increased the mRNA levels and protein expression of TRPC3, and via TRPC3 increased cellular proliferation and migration [54]. E2 upregulates TRPV1 expression, participating in pain induction, endometriosis and bone resorption. TRPV1 mRNA levels have been shown to be decreased by E2. Through GPR30, E2 modulates TRPV1 phosphorylation to participate in pain sensitization. Through non-genomic effects, E2 has been shown to both potentiate and decrease capsaicin-evoked currents of TRPV1 [55]. Yang et al., found that TRPV6 expression was increased in human endometrium after E2 treatment [56]. Similarly, in mouse uterine tissues, TRPV5 and TRPV6 were upregulated with E2 treatment; bisphenol A (BPA) that has estrogenic effects, also significantly increased TRPV5 and TRPV6 expression but not to the degree of E2 [57]. Upregulation of TRPA1 by E2 participates in the pathophysiology of endometriosis. Furthermore, through non-genomic effects, E2 increments TRPA1 activation in glucose-induced insulin secretion [55].

Through these studies, we can observe the diverse interactions that estrogens can have over SOCCs modulation and the repercussions in different pathological states.

### 3.3. Ryanodine Receptor

In the SR, one of the mechanisms in charge of the Ca^2+^ efflux to the cytosol is the RyR. In mammals, three isoforms are found -1, -2, and -3; in mouse ASM, RyR1, and RyR2 are the predominant isoforms, with minimal expression of RyR3 [23,24,58]. It has been found that he endogenous modulation of RyR activity in ASM cells is through the CD38/Cyclic ADP-ribose signaling pathway, regulating [Ca^2+^]i. The membrane-bound protein CD38 synthesizes or degrades cADPR, which functions as a ligand for the protein FKBP-12.6 (12.6 kDa FK506-binding protein). In turn, FKBP112.6 binds to the RyR, providing stabilization and reducing the opening probability of the channel. When cADPR binds to the regulatory protein, this causes a conformational change in the RyR, activating Ca^2+^ release to the cytosol [59,60,61]. Alterations in this signaling pathway, such as the upregulation of CD38 induced by TNF-α, IL-1β, and IFN-γ, can lead to an asthmatic phenotype of the ASM cells, characterized by airway hyperresponsiveness (AHR) that might develop when an already enhanced [Ca^2+^]i concentration is increased further after Gq protein coupled receptor (GPCR) activation [59,61]. Until now, no non-genomic effects of estrogens over RyRs in ASM cells have been reported. However, in rat cardiomyocytes, bisphenol A (BPA) and bisphenol S (BPS) (estrogenic endocrine-disrupting chemicals) have shown acute effects over RyR activity [62,63]. At nanomolar concentrations, both substances significantly altered characteristic RyR-mediated SR Ca^2+^ sparks by transiently and rapidly (30s–5 min) increasing the phosphorylation of RyR at the serine 2808 site through protein kinase A(PKA) activity, as well as phospholamban (PLB, a protein that binds to SERCA and regulates its activity) by Ca^2+^/CaM-dependent protein kinase II (CAMKII). This effect was completely abolished when a selective-ERβ blocker was used (PHTPP), but not with a selective-ERα blocker (MPP), indicating that this effect is dependent on the ERβ signaling pathway [62,63].

In mouse ASM cells, long term exposure (24–48 h) to E2 (at physiological levels) was shown to upregulate the expression of CD38 (Figure 3) (Table 1) [64]. As mentioned before, CD38 is a key component in the mechanisms in charge of Ca^2+^ homeostasis by regulating the activity of RyRs. The genomic effect that estrogens could have directly on RyR expression in ASM cells has yet to be explored. In other models, estrogens have shown to impact RyR expression. In uterine arteries from pregnant sheep (a period of higher estrogenic levels), all three isoforms of RyR were upregulated, causing an increase in Ca^2+^ sparks. Additionally, in uterine arteries from nonpregnant sheep treated ex vivo with estrogen and progesterone mimicking pregnancy conditions, a similar upregulation of RyR was observed [65]. In female rat cardiomyocytes, RyR2 is expressed at higher levels than in male rats [66,67,68]. In contrast, other reports state that in female cardiomyocytes, RyR2 phosphorylation by CaMKII/PKA is reduced, causing lower Ca^2+^ sparks [66,69,70]. Another possible mechanism of modulation through the ER signaling pathway is by direct protein–protein interaction. Recently, it was discovered that ERβ has an atypical non-genomic effect over the RyR, in the neuronal cell line HT-22. In these cells, RyR2 and ERβ have varying levels of co-localization, and in electrophysiological studies using RyRs from mouse brain incorporated into artificial lipid bilayers, the application of unliganded (E2-free) ERβ1 monomers caused a significant increase in single-channel currents under basal [Ca^2+^]i of 100 nM. This effect caused by the addition of ERβ1 could indicate a synergic interaction with Ca^2+^ and RyR that increases the open probability of the channel and could potentiate RyR activity of Ca^2+^-induced Ca^2+^-release [71]. The modulatory activity that the complex E2-ERβ1, or the other isoforms of the ERβ and ERα, could have on the RyR remains to be determined. It might be possible that RyR-ERs interact in the ASM. This probability warrants further and more precise studies.

### 3.4. IP_3_ Receptor

The other mechanism that releases Ca^2+^ from the SR is the IP_3_R, activated by its agonist IP_3_ (inositol 1,4,5-trisphosphate) [23,24,37]. As extensively known, when an agonist binds to a membrane GPCRs, phospholipase Cβ (PLCβ) is activated and hydrolyzes the lipid phosphatidylinositol 4,5-biphosphate, generating IP_3_ and diacylglycerol (DAG) [23,24,37]. In mammals, three isoforms of the IP_3_R are expressed (IP_3_R1, -2, and -3); all isoforms have been identified in ASM (Figure 3) [24,72,73,74,75]. Afterward, IP_3_ binds to the IP_3_R in the SR, generating the release of Ca^2+^ from these internal stores. This pathway has usually been implicated in the agonist-induced contraction of the ASM. However, IP_3_R participation in the bronchodilation mechanism induced by the TAS2R pathway (Type 2 taste receptor) has also been proposed [76,77].

The genomic and non-genomic effects that estrogens could have on the IP_3_ signaling pathway in ASM cells remain to be explored, but estrogen modulation in other cell types has been reported (Figure 3). E2 (1 nM) increased IP_3_ production after 6 h of exposure (non-genomic effect) in rat oviduct smooth muscle cells, a phenomenon mediated through an increase in PLC activity [78]. In HEPG2 cells (human hepatoma cell line), the addition of E2 (nanomolar range) induced a rapid increase in IP_3_ production [79], which was also observed in female rat chondrocytes [80]. Treatment of rat osteoblasts with E2 (100 pM) caused rapid transient increases in [Ca^2+^]i via the PLCβ-IP_3_ pathway [81]. E2 treatment has been shown to cause rapid increases in [Ca^2+^]i through an ER interaction with type 1a metabotropic glutamate receptors (mGluR1a), activating the PLCβ-IP_3_ pathway in female rat astrocytes [82]. Among the genomic effects, it has been reported that IP_3_R1 expression is suppressed after 48h exposure to E2 (10 nM) in human g-292 osteosarcoma cells and rat osteoblasts [83]. In rat choroidal plexus epithelial cells, E2 (nanomolar range) downregulated the expression of receptors TAS2R109 and -144, as well as PLCβ2, resulting in a decrease in [Ca^2+^]i in response to TAS2R agonists [77,84].

### 3.5. Na^+^/Ca^2+^ Exchanger

During the membrane depolarization phase in ASM, an accumulation of Na^+^ underneath the plasma membrane takes place, influencing the Ca^2+^ homeostasis through the generation of local concentration gradients. The consequent modulation of the [Ca^2+^]i through the Na^+^ gradients results in various physiological processes, depending on the magnitude, time and region, including contraction, proliferation, protein synthesis and apoptosis, among others [85].

The NCX serves as one of the [Ca^2+^]i buffer mechanisms by extruding Ca^2+^ from the cytosol to the extracellular space [23,24]. The NCX carries three Na^+^ ions into the cytosol while extruding one Ca^2+^; three isoforms, NCX1, -2, and -3 are known, and the most prominent isoform in ASM is the variant NCX1.3 [24,86,87,88,89]. The participation of NCX in Ca^2+^ homeostasis in ASM seems to be minor (Figure 3) [24,90], and apparently its reverse mode (NCX_rev_) has higher importance in ASM physiology. Interestingly, NCX_rev_ introduces Ca^2+^ and extrudes Na^+^ [24]. NCX_rev_ plays a preponderant role in agonist-stimulated ASM, as for instance, an inhibitor of NCX_rev_ (KB-R7943, 10 µM) attenuated the [Ca^2+^]i increase and contraction induced by carbachol (CCh 100 µM) [91]. The Ca^2+^ influx caused by the removal of Na^+^ in hASM and mouse ASM was blocked by KB-R7943 [91]. 

Seemingly, NCX_rev_ also has an important function during oscillatory contractions in mouse ASM induced by potassium channel blockade with tetraethylammonium chloride (TEA). In this experiment, two pattern changes of [Ca^2+^]i were induced. One was a high-frequency oscillation, and the other a low-frequency rhythmic oscillation. Both types of Ca^2+^ changes participate in triggering ASM contraction, and they might participate in other physiological processes in the ASM. Remarkably, these oscillations augment [Ca^2+^]i, an increase that activates NCX, initiating the relaxation phase by extruding Ca^2+^ [92]. Alterations of the NCX also participate in pathological conditions; TNF-α or IL-13 treatment of hASM cells upregulated the expression of NCX1, and the treatment with KB-R7943 abolished methacholine induced AHR in an allergic mouse model [91,93]. This was also observed in a chronic allergen-induced AHR murine model, where NCX1 was upregulated and had higher NCX_rev_ activity [94]. 

The effects of E2 on NCX modulation have not been as extensively investigated as other mechanisms. The acute exposure to E2 in hASM cells did not have a significant effect on NCX (Table 1) [20], and the effects of a chronic exposure to E2 on NCX in ASM cells has not been explored yet (Figure 3). Likewise, the participation of NCX in other tissues’ physiology is inconclusive. The chronic treatment with E2 (3 days) in prepubescent female rats resulted in downregulation of NCX1 expression in the esophagus [95]. In female rat cardiomyocytes, OVX (ovariectomy) caused downregulation of NCX expression that was reversed by E2 treatment [66,96]. Similarly, in female rabbit cardiomyocytes, NCX expression was greater than in males [66,97,98], and cardiomyocytes incubated with E2 1 nM during 24 h presented a 50% increase in NCX1 protein expression and I_NCX_ density mediated by ERs [98]. Contrary to these findings, OVX or E2 replacement therapy did not alter NCX expression in rat cardiomyocytes. However, NCX activity was significantly increased after OVX in a protein kinase A (PKA)-dependent way; this effect was reverted after E2 treatment [66,99]. Likewise, in cardiomyocytes from OVX guinea pigs, NCX activity was increased by 20%, and this effect was reverted with E2 treatment [66,100]. Moreover, E2 supplementation has been found to have a cardioprotective effect in ischemia/reperfusion injury models [101].

In this sense, NCX overexpression models after myocardial infarction (MI) caused an overload in [Ca^2+^]i in male mice but not in female mice. Male mice cardiomyocytes, when exposed to E2 (nanomolar range), decreased [Ca^2+^]i post-MI in a concentration-dependent manner [101]. Interestingly, in a group of female transgenic mice, the NCX overexpression did not lead to [Ca^2+^]i overload, indicating the protective role of E2 to compensate for the greater activity of NCX [101]. These findings point out that E2 exerts a protective function in cardiac myocytes. Indeed, further research clarified that post-MI E2-confered protection was mediated by NCX. In another study, it was confirmed that the myocardial contractile function (left ventricular developed pressure, dP/dt_max_, dP/dt_min_) in male transgenic NCX overexpression mice was significantly higher than in their WT counterparts, as well as in OVX transgenic females, but not in the transgenic or SHAM transgenic female groups. These results implicate NCX in both the contractile and relaxation aspects of the heartbeat. In the post-MI/reperfusion injury phase, the function recovery in transgenic males was lower than in WT males; however, female WT and transgenic NCX mice had a similar recovery, compared with the significantly diminished recovery in the OVX group. This post-ischemic functional recovery pattern coincides with the lower recovery of energy metabolites (ATP and phosphocreatine) as well as the alternans (heartbeats of alternating large and small amplitude at equal intervals) observed only in the hearts of the male transgenic and female OVX transgenic mice, corresponding to [Ca^2+^]i overload. These findings suggest a protective role of estrogen over the NCX activity during ischemic/reperfusion injury [102].

Furthermore, E2′s protective capacities were also observed in neurons, where nanomolar concentrations of E2 exerted rapid effects over the NCX function, increasing the outward Ca^2+^ current and decreasing the Ca^2+^ influx mediated by NCX; this effect was potentiated by insulin-like growth factor 1 (IGF-1) [103]. These effects seem to be independent of the canonical estrogen signaling pathways, since the presence of an inhibitor of estrogen receptors (ICI182780, 10 µM) did not alter the results. The non-genomic activity of E2 over NCX in neurons leads to maintaining [Ca^2+^]i at lower levels, preventing the activation of Ca^2+^-dependent apoptosis. These mechanisms could be especially useful to counteract the cytotoxicity induced by glutamate through NMDA or AMPA receptor activation [103,104].

### 3.6. Plasma Membrane Ca^2+^ ATPase

To restore [Ca^2+^]i to basal levels after increases induced by agonist stimulation, Ca^2+^ can be pumped across the membrane and out of the cell against its electrochemical gradient, expending ATP during the process [23,24,37,105]. This mechanism is achieved by the ASM plasma membrane ATPase (PMCA) that has four different isoforms: PMCA1, -2, -3 and -4, but only PMCA1 and PMCA4 are expressed in this tissue (Figure 3) [24,105]. PMCA activity in ASM cells has been implicated in many intracellular processes, including contraction regulation [24], cellular proliferation [105], and even apoptosis [105]. It is possible that dysfunction of this protein could lead to an increase in [Ca^2+^]i and favor AHR [24,105].

Modulation of PMCA in ASM cells by estrogen, either through genomic or non-genomic actions, has not been described (Figure 3). However, this phenomenon has been studied in other cellular types. For instance, in prepubescent female rats treated for 3 days with E2 (40 µg/kg/day), the PMCA1 expression in the esophagus was decreased [95]. In MCF-7 cells (breast cancer cell line), PMCA4b isoform expression was increased by E2 (1 nM) treatment in a way mediated by ERα [106]. Conversely, E2 treatment for 24 h decreased the expression of the isoforms PMCA2 and -4 in human fibroblast-like synovial cells (HFLS) and in mouse macrophage-like cells in a dose-dependent manner [107]. In another study, E2 exposure for 24 h did not alter PMCA expression in distal tubule kidney cells; however, PMCA activity was significantly enhanced [108]. In human endometrium, PMCA1 expression was significantly upregulated when treated with E2 (physiological range) for 48 h [56], compared to the decrease of PMCA1 expression in mouse uterus following E2 treatment [57]. E2 has also been described to participate in mechanical pain sensitivity through PMCA2. In female mice, OVX increased mechanical pain sensitivity through PMCA2; this was reversed with E2 supplementation, in a ER-mediated way [109]. It should be noted that alterations in the Ca^2+^ machinery might participate in various pathophysiological processes in accordance with the type of cell implicated.

### 3.7. Sarcoplasmic Reticulum Ca^2+^ ATPase

Another essential Ca^2+^-pump that participates in intracellular Ca^2+^ reuptake that is located on the SR membrane of ASM cells is SERCA. It is in charge of driving Ca^2+^ ions against its electrochemical gradient by ATP consumption to reestablish b[Ca^2+^]i and restoring depleted Ca^2+^ internal stores [24,37,105]. Three isoforms of the SERCA protein are known: SERCA1, -2, and -3, with various alternative splicing isoforms. ASM expresses isoforms SERCA2a and 2b, which predominate [24,110,111]. Alterations in SERCA expression or activity can lead to increases in [Ca^2+^]i, a phenomenon that has been linked to an asthmatic ASM phenotype that contributes to airway remodeling and hyperresponsiveness [24,105,111,112,113].

Interestingly, hASM cells exposed to E2 apparently showed no acute effects on SERCA activity (Figure 3) [20]. Contrastingly, the estrogen-induced genomic effects on SERCA were explored in hASM cells. They were treated with E2 (1 nM), ERα agonist (PPT, 10 nM) or an ERβ agonist (WAY, 10 nM) for 2 hrs, and then incubated with TNF-α or IL-13. The Ca^2+^ response to histamine in the presence of TNF-α or IL-13 was significantly higher compared to the vehicle. A similar response was observed in the E2 and PPT groups, but the effect was reverted in the WAY group, that showed a response similar to the vehicle group. The time of [Ca^2+^]i decay in the response induced by histamine was higher in the group treated with TNF-α or IL-13. This response was reversed by a treatment with WAY but not by E2 or PPT (Figure 3) [35]. These results were attributed to the treatment’s effects on SERCA2 expression, since there was a significant reduction in protein expression in the TNF-α or IL-13 treated cells, which was reverted in WAY treated cells but not in to those administered E2 or PPT (Table 1) [35].

The evidence indicates that [Ca^2+^]i can be modulated by estrogens by the coordinated participation of several target proteins, and that changes in Ca^2+^ availability induced by estrogens is dynamic and additive exerted through genomic and non-genomic events. 

**Table 1 ijms-24-07879-t001:** Summary of the effects of estrogens on the calcium handling mechanisms in the airway smooth muscle.

	*Acute*		*Chronic*	
Calcium Handling Mechanisms	Pathway	Effect	Pathway	Effect
Voltage-dependent Ca^2+^ channels (VDCCs)	ERα	Inhibition [20,34]	ERβ	Inhibition [35]
Store-Operated Calcium Channels (SOCCs)	ERα	Inhibition via STIM1 phosphorylation [20,50]	ERβ	Downregulated STIM1 and Orai1 expression [49]
			ERα	Upregulated STIM1 and Orai1 expression [49]
Ryanodine Receptor (RyR)	Unknown	Unknown	ERs	Upregulates CD38 expression [64]
IP_3_ Receptor (IP_3_R)	Unknown	Unknown	Unknown	Unknown
Na^+^/Ca^2+^ Exchanger (NCX)	No effect	No effect [20]	Unknown	Unknown
Plasma Membrane Ca^2+^ ATPase (PMCA)	Unknown	Unknown	Unknown	Unknown
Sarcoplasmic Reticulum Ca^2+^ ATPase (SERCA)	No effect	No effect	ERβ	Upregulates SERCA2 expression [35]

## 4. Potassium Handling Mechanisms in Airway Smooth Muscle and Their Modulation by Estrogens

Airway basal tone and diameter represents a balance between constriction and relaxation of the ASM cells that allows adequate airflow through the respiratory tract. Potassium channels in ASM largely determine the membrane’s voltage. K^+^ electric conductance through the cell membrane is responsible for keeping the membrane potential at rest close to the equilibrium potential of K^+^, thus playing an essential role as modulator of ASM homeostasis [114,115]. On the other hand, ionic currents through K^+^ channels allow smooth muscle hyperpolarization, contributing to ASM relaxation. In fact, this is the mechanism of action of many bronchodilators as, for instance, β_2_-adrenergic agonists, the main bronchodilators used in the therapy of asthma [114]. The family of potassium channels is incredibly varied, but the most significant in ASM are high-conductance Ca^2+^-activated potassium channels (maxi-K, or BK_Ca_), voltage-activated potassium channels (Kv), and ATP-regulated potassium channels (K_ATP_), although the latter channels have been identified in ASM cells, they do not seem to play a significant role in regulating airway contraction or relaxation [114,115,116,117,118]. To maintain the membrane potential at rest, ASM requires the participation of the Na^+^/K^+^ ATPase (NKA) [119,120], and some exchangers, such as the Na^+^/K^+^/Cl^−^ cotransporter (NKCC) (Figure 4) [121].

### 4.1. Ca^2+^-Activated K^+^ Channels

The Ca^2+^-activated K^+^ channels (K_Ca_) are constituted by three families: the large-conductance Ca^2+^-activated K^+^ channels (BK_Ca_), the intermediate conductance (IK_Ca_), and the small conductance (SK_Ca_). BK_Ca_ have the highest importance in ASM cells [114]. In a cell at rest, Ca^2+^ ions that might activate BK_Ca_ usually are the result of Ca^2+^ sparks released from the SR that, after binding to the channel, raise the open-state probability of the channel (Figure 4) [114].

As already mentioned, β_2_-adrenergic agonists are the main therapeutic option for bronchodilation (in asthma and chronic obstructive pulmonary disease [COPD]), and BK_Ca_ channel activation is one of the key mechanisms of action of these drugs [114,122]. BK_Ca_ channels in ASM have also been involved in airway inflammation, as it has been reported that IL-4 induces rapid, large increases in the channel activity, which was reverted by the presence of IL-13 [123]. Many allergic mouse models display AHR development and maintenance mediated by the Th2-mediated cytokine pathway, particularly IL-13 and IL-4 [124]. Meanwhile, the intermediate conductance K^+^ channel IK_Ca_3.1 was identified to be involved in ASM cell proliferation, migration, and regulation of the expression of contractile phenotypic marker proteins (smooth muscle myosin heavy chain, smooth muscle α-actin, and myocardin), and is overexpressed in asthmatic mice [125,126,127]. The IK_Ca_3.1 channel could be a potential therapeutic target for airway remodeling treatment in asthma and COPD.

Interestingly, the non-genomic activity of E2 on BK_Ca_ was examined in mouse ASM tracheal and bronchial rings incubated for 24 hrs in serum containing immunoglobulin E from healthy humans and asthmatic patients. A 30 min pretreatment with E2 (100 nM) attenuated the AHR induced by carbachol in the asthmatic group. The addition of a nonselective E2 receptor antagonist (ICI 182780) abolished this effect [128]. E2 was shown to stimulate the activity of BK_Ca_, an effect that was corroborated through single-cell electrophysiological experiments. It was found to be mediated by the NOS (nitric oxide synthase)-cGMP (guanyl cyclase)-protein kinase G pathway when the addition of a PKG inhibitor (KT5823, 300 nM) attenuated the activation of BK_Ca_ by E2 [128]. Similarly, the effect of rapid modulation of BK_Ca_ channels by E2 was observed in hASM cells, but Seibold et al. demonstrated that it was linked to an African-specific single nucleotide polymorphism (SNP) in the KCNMB1 gene [129]. The SNP C818T of the KCNMB1 gene, which codes for the β1 subunit of the BK_Ca_ channel, is associated with reduced activation of the channel, leading to clinically significant functional impairment in male patients. On the contrary, in women, this effect was not observed; E2 can induce phosphorylation of the channel via NO/cGMP/PKG pathway to compensate for the SNP defect [129]. Meanwhile, in rat vascular smooth muscle cells, the acute application of E2, at micromolar concentrations, activates BK_Ca_ inducing vasorelaxation [130]. Similarly, in xenopus oocytes, E2 evokes rapid effects over BK_Ca_, activating the channel, perhaps through direct binding [131]. 

However, the genomic effects of estrogens over BK_Ca_ in ASM have not been explored as has been the case in other tissues. The chronic treatment with E2 in OVX rats did not alter the expression of BK_Ca_ in vascular smooth muscle [36]. Nevertheless, in pregnant sheep uterine arteries, the expression of the β1 subunit of BK_Ca_ was increased, and myocytes of non-pregnant sheep treated with E2 reach a similar effect, leading to greater channel activity [132]. Through genomic effects, chronic incubation with E2 increased BK_Ca_ expression via ERβ in human and mouse neuroblastoma cell lines [133]. The same effect was observed in GT1-7 cells (gonadotropin-releasing hormone neuronal cell line) where three days of treatment with E2 at physiological levels induced an increase in BK_Ca_ currents mediated by ERβ, attributed to an increase in expression of the α- and β4-subunits of the BK_Ca_ channel [134]. In human uterine smooth muscle, E2 decreased the expression of BK_Ca_, and reverted the altered expression of the channel in adenomyosis cases [135].

In MCF-7 cells, at physiological levels, E2 regulates cell proliferation in a dose-dependent manner by activating BK_Ca_ independently of ERs [136,137]. BK_Ca_ expression appears to be upregulated by E2 treatment in mice cervical cancer cells, which are estrogen-sensitive cancer cells. [136,138]. The endocrine-disrupting chemicals BPA, BPS, and BPF all demonstrated to regulate the expression of BK_Ca_, diminishing it after 48 h through the activation of the ERβ pathway [139,140].

The microRNA (small non-coding RNA that regulate gene expression by repression or degradation of the mRNA) miR-16-5p has the potential to modulate the expression of IK_Ca_3.1 [141,142], and E2 was shown to suppress miR-16 in MCF-7 cells via ERα [141,143], indicating a possible route through which E2 could regulate IK_Ca_3.1 and cell proliferation in ASM cells. Indeed, it was demonstrated that, in asthmatic individuals, miR-16 was upregulated, and, through functional validation, resulted in reduced ASM cell growth [144]. Meanwhile, in another study, miR-16 was also associated with β-agonist resistance and suppression of β_2_ adrenergic receptors [145]. It would be interesting to explore how E2 (possibly via miR-16) modulates IK_Ca_3.1 in ASM cells.

### 4.2. Voltage-Activated K^+^ Channels

Voltage-activated K^+^ channels (Kv) represent a crucial class of K^+^ channels that regulate muscle tone by controlling resting membrane potential. They are also determinants in airway smooth muscle excitability and represent a significant target for different modulators and drugs that negatively regulate bronchoconstriction [115]. In mammals, 40 genes encode for the Kv channels, each gene corresponds to the α subunit. Kv channels are homotetramers of identical or similar α subunits, sometimes they can also contain auxiliary β subunits, adding to the wide diversity of this family (Figure 4) [146]. The Kv channels can be classified into 12 different subfamilies in accordance with their hydrophobic domain containing the six transmembrane segments [146]. Although all the K^+^ channels share a similarity in the selectivity filter (SF) in the pore structure, the greatest divergence arises in the different gating mechanisms. In the Kv family, the channels are activated by the changes in the electric field built across the excitable cell membrane. The channel is activated by detecting changes in the membrane’s voltage through its voltage-sensing domain (VSD), inducing a conformational change, propagated by a helical linker, that produces the opening of the pore and allows for the efflux of K^+^ [146]. In ASM cells, various channels are present, including three channels from the shaker family (Kv1.1, Kv1.2, and Kv1.5), as well as the Kv7 channel [115,117]. Kv7 is present in guinea pig and human ASM cells, and through electrophysiological studies, it has been confirmed to contribute significantly to the maintenance of the resting membrane potential, airway diameter, and AHR regulation, although in mice, it did not appear to significantly impact membrane voltage or contraction by muscarinic agonists [117,147].

Regarding the influence that estrogens play on these channels, it was published that the expression of the Kv1.5 channel subtype in vascular smooth muscle cells was decreased in OVX rats, and this effect was reversed with the treatment of E2 or tamoxifen [36]. Contrastingly, in OVX rabbit hearts, chronic E2 treatment downregulated the expression of Kv1.5 and Kv7.1 [148]. E2 treatment augmented the expression of Kv7.5, but not Kv7.2 and 7.3, in OVX guinea pig neurons [149,150], and in mouse pancreatic β cells, BPA chronic treatment downregulated the expression of the KV2.1/2.2 channels [139].

The E2 treatment of HEK 293B cells (48 h) enhanced the activity of Kv11.1 channels but did not alter the expression of the channel. Instead, post-transcriptional modifications boosted the interaction between Kv11.1 and the chaperone proteins Hsp90 and Hsc70, improving trafficking of the channel to the membrane [136,151]. The hormonal regulation of the KCNH2 gene trafficking causes a shortening of the QT interval and could serve as a potential therapeutic target for patients at risk of presenting prolonged repolarization [136,151] In rat arterial smooth muscle, E2, at supraphysiological levels, inhibited Kv channel inhibition induced by serotonin independently of the ERs, via Src protein [136,152]. E2 could also modulate cellular proliferation through Kv10.1, in HeLa cells (cervical cancer cell line) E2 at picomolar levels upregulates the expression of Kv10.1 mediated by ERα [136,153,154]. Even though the effect that estrogens could exert over the Kv channels in ASM cells remains unknown, in other cellular models it has been defined that they participate both in physiological and pathological processes, and the same could be the case in ASM.

### 4.3. Na^+^/K^+^ ATPase

In ASM, the NKA is known to contribute to resting membrane potential regulation and maintenance of muscle tone. Its participation in spontaneous phasic activity during contraction has been recognized as well [119,120]. The ionic exchange of NKA extrudes 3 Na^+^ molecules and pumps in 2 K^+^ into the cytosol for every ATP used; it is an electrogenic pump that promotes a Na^+^ outward current (Figure 4) [119]. During oscillatory contractions in mouse ASM induced by TEA, NKA has been identified to participate in the relaxation phase of the oscillations. When activated, NKA reduces the differential potential of the cellular membrane, inactivating LVDCCs and causing relaxation [92]. It should be mentioned that TEA is a drug used to induce ASM contraction by depolarizing the membrane through the inhibition of K^+^ channels. When activated by K^+^, the pump induces muscle relaxation by hyperpolarizing the membrane. In accordance, inhibiting the ASM NKA pump with ouabain (100 µM) produces contraction [120,155,156,157]. The NKA also plays a role in AHR. In guinea pig ASM cells, bradykinin (an agonist of the bradykinin GPCR receptors, known to cause bronchoconstriction in asthmatics) increased the NKA activity mediated by the activation of the B2 receptors and stimulation of Na^+^ influx through the Na^+^-H^+^ exchanger (NHX) [120]. NKA blockade with ouabain can alter ASM tone and induce muscle contraction by favoring the influx of Ca^2+^ through the VDCCs and NCX [85,120]. The differential regulation presented by the increase of NKA activity by bradykinin could lead to a negative feedback mechanism that opposes the contraction usually elicited by B2 receptor stimulation [120].

The effects of E2 on NKA activity has not been explored in ASM cells. In H9C2 rat cardiomyocytes, E2 enhanced the expression of NKA in a concentration-dependent manner [158,159]. In male rat cardiomyocytes, NKA activity, expression, and phosphorylation (possibly through Akt and ERK 1,2 regulation) were enhanced after 24 h E2 treatment [160]. A similar effect was observed in HSG and HeLa cells, where E2 treatment, through ERβ activation, upregulated the expression of the β1-subunit of NKA, and through the N-myc downstream-regulated Gene 2 (NDGR2) the degradation of the β1-subunit was decremented, enhancing NKA activity [158,161]. Moreover, in erythrocytes from women, NKA activity was enhanced when E2 levels were higher [158,162]. In female rat aorta, the activity of NKA is increased through NO, and an upregulation of the α2-isoform of NKA was noticed as well. Inversely, OVX blunted the effect [163]. Contrary to this, in another study, OVX increased the activity of NKA in rat vascular smooth muscle [164].

### 4.4. Na^+^/K^+^/Cl^−^ Cotransporter

The Na^+^/K^+^/Cl^−^ cotransporter (NKCC) facilitates the movement of Na^+^, K^+^, and Cl^−^ ions across the membrane in an electrically neutral way (Figure 4) [121]. It is expressed in ASM cells, and participates in different processes, including ASM cell proliferation, confirmed by its inhibition through loop diuretics treatment (bumetanide and furosemide) [165]. NKCC1 immunoreactivity was found in airway epithelial cells and alveolar type II cells, and in mice sensitized with ovalbumin (OVA) overexpression was seen. The inhibition of the cotransporter with furosemide reduced overall airway responsiveness and reverted the AHR induced by OVA sensitization [166].

The effect of E2 over the NKCC has been studied in other cell types. E2, at nanomolar concentrations, has a neuroprotective effect by decreasing edema formation after cerebral ischemia in rats by attenuating the NKCC activity [167]. After 24 h treatment with E2 (1 nM), NKCC expression decreased in bovine cerebral microvascular endothelial cells. Nevertheless, when endothelial cells were subjected to shear stress, E2 did not alter NKCC expression [168]. Contrary to this, in OVX rat hippocampus, the expression of NKCC decreased, and 24 h treatment with E2 reverted this effect [169]. In developing rat hypothalamus neurons, E2 treatment for 24 h significantly enhanced NKCC1 phosphorylation by increasing the expression of the kinases SPAK and OSR1 but did not modify the protein expression levels [170].

In rat aorta, the NKCC exchanger seems to participate in phenylephrine (PE) contraction in a gender-specific manner. In female rats, inhibition of NKCC1 did not modify the maximal contraction with PE, while OVX increased NKCC1 activity and E2 treatment blunted this effect [171]. The modulatory effect of E2 over the NKCC in ASM cells is still unknown but considering the significance that it has in the regulation of physiological and pathological processes in other cells model, it would be interesting to explore in the future the role it plays in the ASM.

## 5. Sodium Handling in Airway Smooth Muscle and Its Modulation by Estrogen

To initiate bronchoconstriction, an increase in [Ca^2+^]i is necessary, although it is not the only contributor to ASM contraction regulation. In the extracellular space, Na^+^ concentration is the highest (140 mM), while the intracellular concentration is normally much lower, about 4–16 mM. This electrochemical gradient is usually utilized by excitable cells to generate action potentials and to facilitate the transport of energetically unfavorable solutes coupled to Na^+^ [172]. In ASM cells, the membranal depolarization initiates contraction, and no single mechanism is in charge of maintaining the equilibrium of the membrane potential. Instead, many membrane mechanisms interact to balance it, including Na^+^ channels that accelerate the rate of depolarization by influencing Ca^2+^ handling mechanisms [85]. As with other ion handling mechanisms, various proteins regulate the influx and efflux of Na^+^ ions. In ASM cells, the main contributors for Na^+^ influx are the NCX, the voltage-gated Na^+^ channels (VGSC), and non-selective cation channels (NSCC), while Na^+^ is extruded through the NKA [85].

### Voltage-Gated Na^+^ Channels

When bearing in mind excitable cells, such as neurons or myocytes, usually voltage gated sodium channels (VGSCs) are some of the first channels to consider. In ASM cells, VGSCs have already been identified to be functionally expressed in humans and rabbits, including the isoforms Nav1.2, 1.5, and 1.7 [85,173,174,175]. The contributors for the Na^+^ currents observed in ASM cells appear to be Nav1.5 and 1.7 [173,175]. Additionally, oscillatory contractions in mouse ASM, Nav channels participate in the enhancement of the depolarization by activating LVDCCs; interestingly, when Nav were inhibited, the contractions disappeared [92]. Furthermore, the participation of these channels could be involved in pathological conditions, the increase in intracellular Na^+^ concentration ([Na^+^]i) activates the NCX_REV_, increasing [Ca^2+^]i (Figure 4) [24,85,176]. Hence, the enhanced *I*_Na_ could participate in the dysfunction of cell contraction, excitation, and remodeling [176]. The chronic treatment of ASM cells with dexamethasone (1 µM for 24 h) diminished the expression of Nav1.7 channel and could prove beneficial in the detrimental mechanisms elicited by these channels in some pathologies [176]. 

Voltage-gated sodium channels also participated in pathologies like cancer, and it has been found that VGSCs are upregulated in these illnesses where they are indispensable for various pathophysiological mechanisms such as metastasis [177]. It is noteworthy to mention that in human breast cancer cells, the non-genomic effects of E2 through GPR30 increased the neonatal splice form of Nav1.5’s (nNav1.5) activity, and this mechanism was crucial for cell adhesion and metastasis [178]. Moreover, E2 has been shown to exert non-genomic cardioprotective effects via Nav modulation. In cardiomyocytes from human inducible pluripotent stem cells (iPSC-CMs) and Chinese hamster ovarian (CHO) cells, E2 (micromolar concentrations) did not elicit changes by itself in Nav currents; however, it significantly reduced the high glucose or inflammatory mediators (bradykinin, histamine, serotonin, and ATP)-induced increments in Na+ currents that caused hyperexcitability and resulted in long QT3 arrhythmias, this effect was mediated by PKC/PKA activation [179].

E2 actions on VGSCs seemingly also participate in pain modulation. In adult dorsal root ganglion (DRG) neurons, ERs differentially regulate Nav expression [180]. In ERα knock out (ErαKO) and Erβ knock out (ERβKO) mice, the expression of Nav1.1, 1.7, 1.8, and 1.9 are upregulated, but in αERKO, the expression of Nav1.6 was decreased but not in βERKO [180]. Opposite to this, E2 treatment increased the expression of Nav1.7 in OVX rat trigeminal ganglions in a dose-dependent manner [181]. Meanwhile, a single acute E2 dose was proven to also modulate Nav activity. In patch-clamp studies, E2 at 10 nM increased Na+ inward currents in rat hypothalamic neurons [182]. 

## 6. Chlorine Handling Mechanisms in Airway Smooth Muscle and Their Modulation by Estrogen

Many ions participate in controlling the cellular electrochemical balance and its membrane potential, and Cl^−^ is no exception [183,184]. Intracellular concentrations of Cl^−^ are approximately 30–50 nM, in turn, the equilibrium potential of Cl^−^ ranges between −30 and −20 mV, more positive than the ASM resting membrane potential, which is around -60 mV. In an ASM at rest, Cl^−^ efflux will lead to membrane depolarization, but when the membrane depolarization is already in course, this Cl^−^ current will contribute to maintaining and enhancing the initial depolarization. Because of these mechanisms, Ca^2+^ enters through LVDCCs activation and a subsequently initiates an ASM contraction (Figure 4) [183,184,185,186]. Cl^−^ homeostasis is maintained through the coordinated activities of a series of proteins in charge of the influx and efflux of this ion through the membrane. It has been established that in ASM, the NKCC as well as several Cl^−^ channels participate in this anion’s management. In fact, the following Cl^−^ channels have been identified in this cell type: Ca^2+^-activated Cl^−^ channels (CACCs), ligand-gated Cl^−^ channels (gamma amino butyric acid receptor, GABA), and the cystic fibrosis transmembrane conductance regulator (CFTR) (Figure 4) [183,184].

Regarding smooth muscle physiology, Cl^−^ channels seem to effectively induce relaxation from a pre-contracted ASM, but apparently have minimal effects on basal tension and even might prevent contraction development [183,187]. This is probably related to the ASM Cl^−^ channels’ localization, and the impact of the channel blockade on Ca^2+^ regulation at the membrane and SR level [187]. The blockade of the Cl^−^ channels potentiates the relaxation induced by β-agonists, especially when combining a Cl^−^ channel blocker with bumetanide (an NKCC blocker), and certainly this effect could be of therapeutic interest in the treatment of asthma [187]. Cl^−^ channel activity also participates closely in Ca^2+^ handling and contraction, not only through Cl^−^ channels present in the plasma membrane, but on the SR. In the SR, Ca^2+^ release is an electrogenic process, and modifications in charges (positive or negative) across the membrane will hinder this process. If the SR Cl^−^ fluxes are inhibited, Ca^2+^ sequestration and Ca^2+^ release induced by agonists will be reduced [183,188]. In ASM oscillatory contractions, Cl^−^ channels play a crucial role, especially in the rhythmic transitions of the membrane potential; this effect is clearly observed in the contraction induced by TEA (a K^+^ channel blocker) [92,183]. During these oscillations, the [Ca^2+^]i increase induced by LVDCCs currents activate CACCs, and Cl^−^ ions participate in the contractile phase by favoring membrane depolarization and enhancing the contraction by further activating LVDCCs [92].

### 6.1. Ca^2+^ activated Cl^−^ Channels

CACCs have been shown to be present in ASM cells, and are activated by the increase in [Ca^2+^]i by activation of LVDCCs or Ca^2+^ release from the internal stores [185,186,188]. The participation of CACCs can be observed in various processes in ASM cells, including modulation of contraction–relaxation, maintenance of muscle tone and membrane potential, and membrane depolarization (Figure 4) [183,185,186,188,189]. The blockade of CACCs also potentiates the relaxant effect induced by β-agonists and could be of interest in the treatment of asthma [187]. One of the most important CACCs identified in ASM cells is TMEM16A (transmembrane protein with unknown function 16, Ano1); seemingly, this channel is involved in ASM tone modulation, and regulates the contraction induced by cholinergic agonists [190,191]. The participation of TMEM26A has also been observed in the relaxation by activation of the odorant pathway via the GPCR OR2W3, in which SOCCs activation initiates a tradeoff between CFTR in the SR and TMEM16A in the plasma membrane to induce relaxation [192].

Notwithstanding, no report exists on the estrogen’s effect on CACCs in ASM cells. However, in HEK293T cells, estrogens inhibit TMEM16A. This inhibitory non-genomic effect of estrogen was observed with E2, E3, and estetrol (E4) at micromolar levels, with the most potent effect observed with E3 [193]. In female CF (cystic fibrosis) patients, periods of E2 plasmatic levels elevation during the menstrual cycle coincided with the reduction of CACCs activity in airway epithelial cells, which caused subsequent mucus plugging. Tamoxifen treatment prevented this effect, and it even enhanced the Cl- currents [194]. In human bronchial epithelial cells, the acute application of E2 (10 nM) reduced ATP-induced CACC currents, but this effect was reverted with tamoxifen; in fact, the CACC currents were potentiated [195]. This estrogenic mechanism could be useful in the treatment of CF patients. This non-genomic effect of estrogens should be explored in ASM cells.

### 6.2. Cystic Fibrosis Transmembrane Conductance Regulator

In ASM cells, the CFTR channel is functionally expressed and plays a role in regulating muscle tone (Figure 4) [196,197,198,199]. In this sense, when compared to the WT littermates, FVB/N ΔF508 mice (deletion of the 508 codon) presented AHR with increased airway resistance against methacholine challenge. This increment in airway resistance developed in the absence of overt lung inflammation or an increase in ASM mass and was attributed to the altered airway mechanics [200]. A loss of CFTR, such as is the case in CF, leads to an “asthma-like” phenotype in patients that present AHR, and is attributed to be a consequence of a delayed Ca^2+^ reuptake and increased myosin light chain phosphorylation after cholinergic stimulation [197,198]. In addition, the CFTR channel and the TMEM16A jointly participate in the relaxation signaling pathway initiated by the odorant receptor OR2W3 in ASM cells [201].

In female CF patients, lung function was measured in correspondence to their menstrual cycle, and it was observed that the forced expiratory volume during one second and the forced vital capacity were significantly higher during the luteal phase when compared to ovulation and menstruation; this could be due to the higher estrogen levels, but the mechanism that could be at play or if it involves the ASM remains to be investigated [202]. In T84 epithelial cells, estrogen seems to modulate CFTR activity; E2 (micromolar levels) induced rapid and reversible inhibition of forskolin-stimulated Cl^−^ secretion, and this effect was also observed with other estrogen derivatives, including the stereoisomer 17α-estradiol that does not bind to the ERs, indicating that the genomic effect is through direct interaction of the estrogen with the CFTR protein [203]. In rats, an increment of CFTR expression in the uterus and ovary was observed after induced ovarian hyperstimulation syndrome (OHSS), in which E2 levels were found to be eight times higher than normal levels [204]. This was corroborated by E2 treatment of OVX rats that mimicked the results of OHSS and through measurement of CFTR activity in freshly isolated uterine epithelia [204]. In another study, CFTR expression was higher in women than in men in duodenal mucosa cells and in duodenocytes treated with E2 (1 nM) for 12 h, where a significant increase in expression mediated by ERα was observed [205]. This was also corroborated by measuring the activity of CFTR in OVX mice. Forskolin (a CFTR activator) stimulation response was markedly decreased, and this effect was reverted with E2 supplementation [205]. Similarly, peritoneal epithelial cells of rats with induced OHSS showed CFTR upregulation, and this genomic effect was reproduced by E2 treatment in OVX rats [206]. In pancreatic epithelial cells, E2 incubation (3.7 nM for 20 h), did not alter CFTR expression levels, but it inhibited cAMP-activation of CFTR and consequently led to cell volume reduction [207]. CFTR is also present in guinea pig cardiomyocytes, and the acute application of E2 (micromolar range) potentiates the *I*_Cl_ induced by isoprenaline in a concentration-dependent manner, independently of the ERs [208].

### 6.3. GABA-Activated Cl^−^ Channels

Another group of Cl^−^ channels are ionotropic receptors activated by a ligand. In ASM cells, ionotropic GABA_A_ receptors have been identified to be functionally expressed in humans and guinea pigs, and GABA can relax the contraction induced by tachykinin in a concentration-dependent manner (Figure 4) [209]. Contrastingly, the activation of GABA_A_ receptor does not affect ASM basal tone, nor does it seem effective in preventing a contraction, but when applied to a precontracted ASM, it induces relaxation effectively [210]. The GABA_A_ receptors are heteromeric pentamers that can be composed of seven different classes of subunits (α1–6, β1–3, γ1–3, δ, π, ξ, and ρ1–3); the mixed composition of the receptor can affect its responsiveness to allosteric activation, particularly attributed to the α subunit [210]. In hASM cells, from the α subunit class, only α4 and α5 are expressed and integrate a hetero pentameric receptor [210], and selective targeting of this subunit with agonists results in activation of the Cl^−^ currents that generate hyperpolarization and relaxation of the precontracted smooth muscle [210,211]. When sensitized to house dust mite (HDM) antigen, GABA_A_ receptor α4-subunit knockout mice in vivo models develop an increase in airway resistance, lung inflammation, and ASM proliferation, when compared to the WT mice [212]. The selective activation of the α subunit of the GABA_A_ receptor could have bronchodilatory potential in asthma or COPD.

Although no reports have been made concerning the effect of estrogens on the modulation of the GABA_A_ receptor in ASM cells, some effects have been observed in other tissues. In female rat brains, E2 treatment after OVX showed a significant increase of the α2 and the γ1 subunits only in regions that also expressed ERs, suggesting that the effect depends on the ER signaling [213]. The effect was also time-dependent, since there was a greater upregulation after 7 d of treatment compared to only 24 h [213]. Through genomic effects, E2 treatment also increased GABA binding sites in OVX rat brain, as soon as after 3 h of incubation, in a dose-dependent manner [214]. Conversely, another report failed to find a significant alteration in the rat brain expression of GABA_A_ receptor in the OVX vs. OVX + E2 treated groups [215]. E2 regulation of GABA_A_ receptor expression is also relevant in the developmental stages. In neonatal female rats, β-estradiol 3-benzoate (EB) treatment upregulated the expression of extrasynaptic α4/δ subunits but decreased the expression of synaptic α1/α4/γ2 subunits [216].

## 7. Conclusions

In summary, a wide range of evidence from different cell types and animal models indicate that estrogens have a promiscuous nature, being able to interact with a variety of regulatory proteins essential in the homeostasis of intracellular ions. The estrogenic effects are not clearly cut but are instead subject to the conditions and variables that surround them. The effects of estrogens can vary greatly depending on the concentration, time of exposure, type of estrogen, requirement or independence of ER signaling, and even cell type. In this context, the effects of estrogens in some proteins such as IP3 receptors, PMCA, SERCA, NKA, NKCC, VGSCs, and CACCs on ASM cells have not been explored in-depth. There is not a single regulatory protein in which at least one study of E2 modulation is reported. Thus, the necessity still exists to better understand the possible effects that estrogens could have in physiological and pathological conditions in ASM cells. In this sense, it seems that single-cell genomics studies and spatial transcriptomics analysis may be quite useful in the following years to unravel the punctual functions of estrogens, particularly on the surrounding tissues where they are synthesized.

## Figures and Tables

**Figure 1 ijms-24-07879-f001:**
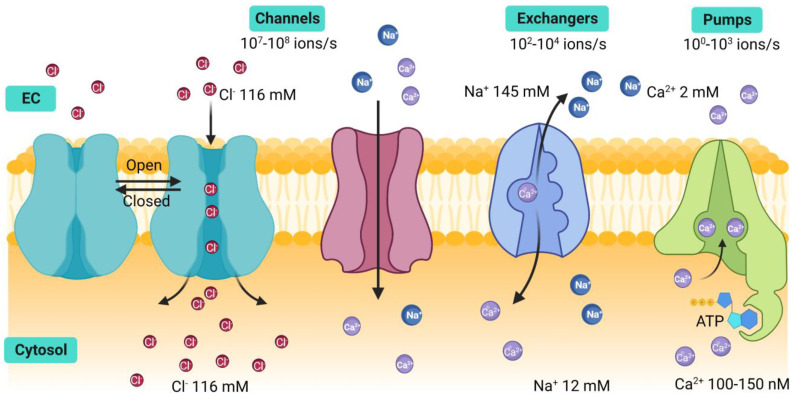
Ion transport proteins. Ion transport proteins are classified into three groups: channels, exchangers, and pumps. These three types of proteins maintain the homeostasis of the intracellular ions either through facilitated or active transport, and changes in the ionic intracellular concentrations can initiate cellular processes. In blue, a chlorine channel is represented; in red, a non-selective cation channel; in purple, the Na^+^/Ca^2+^ exchanger; and in green, the plasma membrane Ca^2+^ ATPase. EC, extracellular; Cl^−^, chloride, Na^+^, sodium; Ca^2+^, calcium; ATP, adenosine triphosphate.

**Figure 2 ijms-24-07879-f002:**
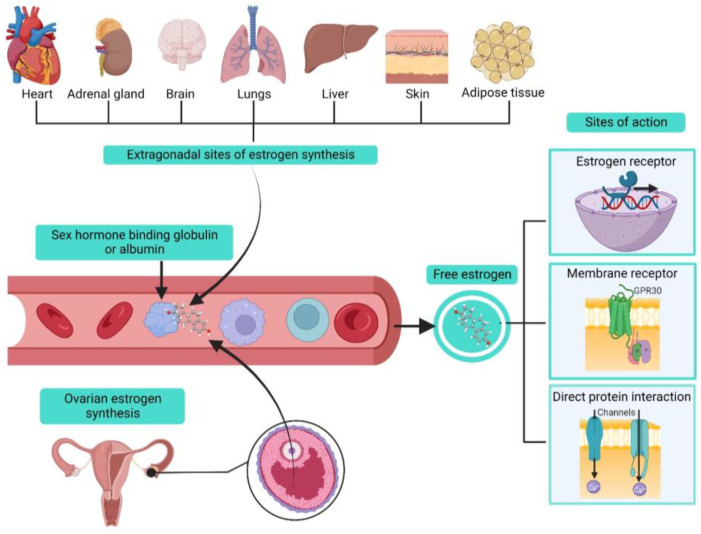
Estrogen biosynthesis and sites of action. The main site of estrogen synthesis are the ovaries. However, several extragonadal synthesis sites exist, and the main requirement for local estrogen production is the tissue expression of the aromatase enzyme. Once the estrogen is synthesized, it can have local activity or it can be liberated into the blood stream, where it will be carried to its action site bound to the sex hormone binding globulin or to albumin. When the estrogen reaches its target, it can produce its effect through three modes of action: through the activation of the estrogen receptors ERα and ERβ, or by interacting with its membrane receptor GPR30 or directly with its target protein. GPR30, G protein-coupled receptor 30.

**Figure 3 ijms-24-07879-f003:**
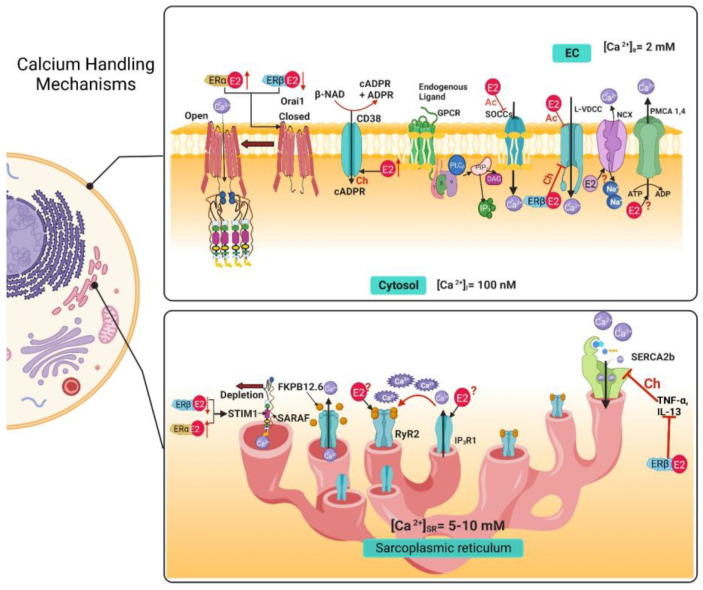
Calcium handling mechanisms in airway smooth muscle and their modulation by estrogens. E2 through non-genomic effects inhibits SOCCs, and the chronic exposure to an ERβ-specific agonist decreases the expression of STIM1 and Orai1, and an ERα-specific agonist increases the expression of STIM1 and Orai1.The genomic effect of E2 increases the expression of CD38, although the non-genomic effect is still unknown. E2 acutely inhibits the LVDCC and the chronic exposure, via ERβ pathway, inhibits this channel. In the sarcoplasmic reticulum, the chronic exposure with an ERβ-specific agonist inhibits the increase in SERCA expression caused by exposure to TNF-α and IL-13. The genomic and non-genomic effects of E2 over RyR2, IP_3_R1, NCX and PMCA are still not reported. EC, extracellular; Ac, Acute; Ch, Chronic; IP_3_, inositol 1,4,5-trisphosphate; [Ca^2+^]i, intracellular calcium concentration; [Ca^2+^]e, extracellular calcium concentration; [Ca^2+^]SR, sarcoplasmic reticulum calcium concentration; E2_,_ 17β estradiol; ERα, estrogen receptor α; ERβ, estrogen receptor β; L-VDCC, L-type voltage dependent Ca^2+^ channel; SOCC, store-operated Ca^2+^ channel, NCX, Na^+^/Ca^2+^ Exchanger; PMCA, plasma membrane Ca^2+^ ATPase; IP_3_R, inositol 1,4,5-trisphosphate receptor; RyR, Ryanodine receptor; SERCA, sarcoplasmic reticulum Ca^2+^ ATPase; FKBP-12.6, 12.6 kDa FK506-binding protein; Orai1; STIM1, stromal interacting molecule; SARAF, Store-operated Ca^2+^ entry-associated regulatory factor.

**Figure 4 ijms-24-07879-f004:**
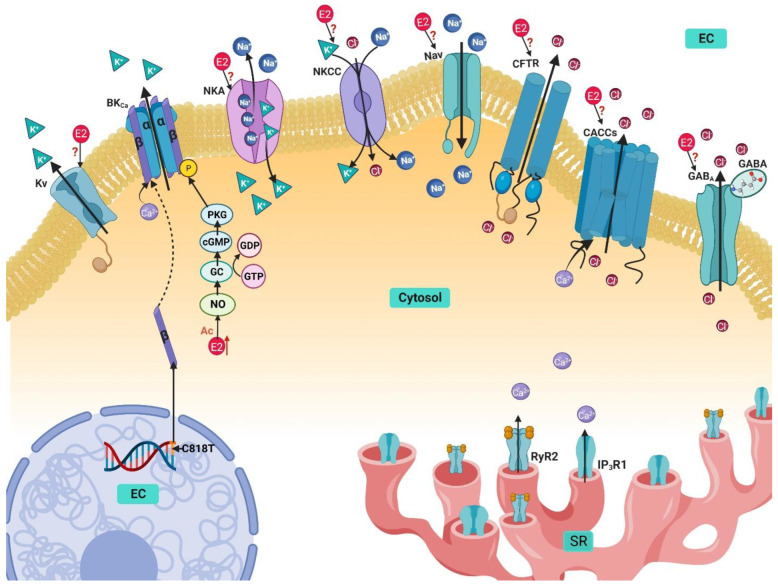
Potassium, sodium and chloride handling mechanisms in airway smooth muscle and their modulation by estrogens. Estrogen acutely increases the BK_Ca_ activity through the NO/GC/cGMP/PKG pathway phosphorylation of the β subunit, and when the SNP C818T is expressed, E2 compensates for the reduced activation of the channel through the activation of the NO/GC/cGMP/PKG pathway. The activity of E2 over the Kv, NKA, NKCC, Nav, CFTR, CACCs, and GABA_A_ receptor proteins in the ASM is still unknown. EC, extracellular; E2, estradiol; Ac, Acute; SR, sarcoplasmic reticulum; Cl^−^, chloride, K^+^, potassium; Na^+^, sodium; Ca^2+^, calcium; P, phosphate; BK_Ca_, high-conductance Ca^2+^-activated K^+^ channel; Kv, voltage-activated potassium channels; NKA, Na^+^/K^+^ ATPase; NKCC, Na^+^/K^+^/Cl^−^ cotransporter; Nav, voltage-gated Na^+^ channels; CACCs, Ca^2+^-activated Cl^−^ channels; CFTR, cystic fibrosis transmembrane conductance regulator; GABA, gamma amino butyric acid; GABA_A_, activated chloride channel; RyR2, ryanodine receptor 2; IP_3_R, inositol 1,4,5-trisphosphate receptor; NO, nitric oxide; GC, guanyl cyclase; cGMP, cyclic guanosine monophosphate; PKG, protein kinase G; GTP, guanosine triphosphate; GDP, guanosine diphosphate.

## Data Availability

Not applicable.

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
