# Peer review of "Estrogenic Modulation of Ionic Channels, Pumps and Exchangers in Airway Smooth Muscle"

_ijms, 2023, doi:10.3390/ijms24097879_

Round 1
Reviewer 1 Report
The review from Romero-Martinez et al. entitled “Estrogenic modulation of ionic channels, pumps and exchangers in airway smooth muscle.” deeply summarizes the role of estrogens in modulating calcium homeostasis in physiological and pathological conditions in ASM cells. The effects of estrogens on the different calcium homeostasis player is well reported highlighting both the well established knowledge and the open questions. The review is well organized and detailed. Seeing the complexity of the topic I have some suggestion to make reading easier and to slim down the text.
1. Some sentences in brachets (e.g. lines 214-219; line 252) could be omitted
2. Some sentences that give an hint but does not go in to details could be removed or better contextualized
Line 105: “in molecular evolution, we can observe…”
Line 193 “various pathological processes of the ASM..” what kind of pathological processes?
3. In line 221 there is a reference to figure 3 that in my opinion is not corrected. Indeed, they are talking about histamine-challenged ASM, but no histamine stimulation has been reported in the figure.
4. Since the text is very rich of information, it would be useful to generate two tables (one for the 3rd paragraph, one for the 4th) in which to summarize the receptors expressed in ASM cells and the effect of the estrogen on these (if known).
Author Response
REVIEWER 1
The review from Romero-Martinez et al. entitled “Estrogenic modulation of ionic channels, pumps and exchangers in airway smooth muscle.” deeply summarizes the role of estrogens in modulating calcium homeostasis in physiological and pathological conditions in ASM cells. The effects of estrogens on the different calcium homeostasis player is well reported highlighting both the well established knowledge and the open questions. The review is well organized and detailed. Seeing the complexity of the topic I have some suggestion to make reading easier and to slim down the text.
- Some sentences in brachets (e.g. lines 214-219; line 252) could be omitted
- Thank you very much for the kind suggestion, we have omitted the mentioned sentences in the brackets on lines 214, 219 and 252
- Some sentences that give an hint but does not go in to details could be removed or better contextualized
Line 105: “in molecular evolution, we can observe…”
Line 193 “various pathological processes of the ASM..” what kind of pathological processes?
A: We thank the reviewer for the advice, we have excluded the statements in line 105 and 193 that weren’t given context.
- In line 221 there is a reference to figure 3 that in my opinion is not corrected. Indeed, they are talking about histamine-challenged ASM, but no histamine stimulation has been reported in the figure.
A: We appreciate the observation, we have changed the location of the reference for figure 3 to Line 215, that better represents the effect depicted in the image.
- Since the text is very rich of information, it would be useful to generate two tables (one for the 3rd paragraph, one for the 4th) in which to summarize the receptors expressed in ASM cells and the effect of the estrogen on these (if known).
A: We have taken into consideration the reviewer´s suggestion and added a table that encompasses the information of both tables requested. The table was added to the new line 535.
|
|
Acute |
Chronic |
||
|
Calcium handling mechanisms |
Pathway |
Effect |
Pathway |
Effect |
|
Voltage-dependent Ca2+ channels (VDCCs) |
ERα |
Inhibition |
ERβ |
Inhibition |
|
Store-Operated Calcium Channels (SOCCs) |
ERα STIM1 phosphorylation |
Inhibition |
ERβ |
downregulated STIM1 and Orai1 expression |
|
ERα |
Upregulated STIM1 and Orai1 expression |
|||
|
Ryanodine Receptor (RyR) |
Unknown |
Unknown |
ERs |
Upregulates CD38 expression |
|
IP3 Receptor (IP3R) |
Unknown |
Unknown |
Unknown |
Unknown |
|
Na+/Ca2+ Exchanger (NCX) |
No effect |
No effect |
Unknown |
Unknown |
|
Plasma Membrane Ca2+ ATPase (PMCA) |
Unknown |
Unknown |
Unknown |
Unknown |
|
Sarcoplasmic Reticulum Ca2+ ATPase (SERCA) |
No effect |
No effect |
ERβ |
Upregulates SERCA2 expression |
Table 1. Summary of the effects of estrogens on the calcium handling mechanisms in the airway smooth muscle.
We thank the reviewers for their comments. They contributed greatly to the improvement of the manuscript.
Cordially,
Dr. Edgar Flores Soto, PhD, corresponding autor. E-mail: edgarfloressoto@yahoo.com.mx

Reviewer 2 Report
This is an interesting review article in which the authors introduced how estrogen regulates ionic channels, pumps and exchangers in airway smooth muscle. Below are my suggestions
1) Fig. 1, the authors only listed a Cl- channel model. Would you be able to include a cation channel? I think this can help readers to understand ion channels in airway smooth muscle better. Optional: may draw a more complex picture to describe the interactions among ion channels and exchangers.
2) The authors made excellent writing for introducing Ca2+ channels, K+ channels, Na channels, and Cl channels. I also notice that the authors mentioned one sentence about TRP channels. Is it good for the author can expand their text to introduce more TRP channels and other non-selective cation channels? I think this is a very important part of the field. So that the reader can understand how estrogen can regulate those channels.
Author Response
REVIEWER 2
This is an interesting review article in which the authors introduced how estrogen regulates ionic channels, pumps and exchangers in airway smooth muscle. Below are my suggestions
1) Fig. 1, the authors only listed a Cl- channel model. Would you be able to include a cation channel? I think this can help readers to understand ion channels in airway smooth muscle better. Optional: may draw a more complex picture to describe the interactions among ion channels and exchangers.
A: We appreciate the observations of the reviewer and have added a cation channel to figure 1 and added the following sentence in the figure´s description in Line 78-79.
“In blue a chlorine channel is represented, in red a non-selective cation channel, in purple the Na+/Ca2+ exchanger, and in green the plasma membrane Ca2+ ATPase.”
2) The authors made excellent writing for introducing Ca2+ channels, K+ channels, Na channels, and Cl channels. I also notice that the authors mentioned one sentence about TRP channels. Is it good for the author can expand their text to introduce more TRP channels and other non-selective cation channels? I think this is a very important part of the field. So that the reader can understand how estrogen can regulate those channels.
- We thank the reviewer´s interest in our work and have added the following text that addresses the reviewer´s suggestion.
Line 260
“ankyrin (TRPA) and melastatin (TRPM),”
Line 310-315
“E2 upregulates TRPV1 expression, participating in pain induction, endometriosis and bone resorption. TRPV1 mRNA levels have been shown to be decreased by E2. Through GPR30, E2 modulates TRPV1 phosphorylation to participate in pain sensitization. Through non-genomic effects, E2 has been shown to both potentiate and decrease capsaicin-evoked currents of TRPV1 [55].”
Line 318-321
“Upregulation of TRPA1 by E2 participates in the pathophysiology of endometriosis. Furthermore, through non-genomic effects, E2 increments TRPA1 activation in glucose-induced insulin secretion [55].”
- Uchida, Y.; Izumizaki, M. Effect of menstrual cycle and female hormones on TRP and TREK channels in modifying thermosensitivity and physiological functions in women. Journal of thermal biology 2021, 100, doi:https://doi.org/10.1016/j.jtherbio.2021.103029.
- Méndez-Reséndiz, K.A.; Enciso-Pablo, Ó.; González-Ramírez, R.; Juárez-Contreras, R.; Rosenbaum, T.; Morales-Lázaro, S.L. Steroids and TRP Channels: A Close Relationship. International Journal of Molecular Sciences 2020, 21, 3819, doi:10.3390/ijms21113819.
We thank the reviewers for their comments. They contributed greatly to the improvement of the manuscript.
Cordially,
Dr. Edgar Flores Soto, PhD, corresponding autor.
E-mail: edgarfloressoto@yahoo.com.mx
